# Multidisciplinary Management of Descending Necrotizing Mediastinitis: Is Thoracoscopic Treatment Feasible?

**DOI:** 10.3390/jcm13082440

**Published:** 2024-04-22

**Authors:** Beatrice Leonardi, Giovanni Natale, Caterina Sagnelli, Antonio Marella, Francesco Leone, Francesca Capasso, Noemi Maria Giorgiano, Davide Gerardo Pica, Rosa Mirra, Vincenzo Di Filippo, Gaetana Messina, Giovanni Vicidomini, Giovanni Motta, Eva Aurora Massimilla, Gaetano Motta, Erino Angelo Rendina, Valentina Peritone, Claudio Andreetti, Alfonso Fiorelli, Antonello Sica

**Affiliations:** 1Thoracic Surgery Unit, University of Campania “Luigi Vanvitelli”, 80131 Naples, Italy; beatrice.leonardi@unicampania.it (B.L.); giovanni.natale@unicampania.it (G.N.); antonio.marella@unicampania.it (A.M.); leonemfg@libero.it (F.L.); francesca.capasso93@gmail.com (F.C.); noemimaria.giorgiano@studenti.unicampania.it (N.M.G.); davide_pica@hotmail.it (D.G.P.); rosa.mirra@studenti.unicampania.it (R.M.); vincenzodifilippo16@gmail.com (V.D.F.); adamessina@virgilio.it (G.M.); giovanni.vicidomini@unicampania.it (G.V.); alfonso.fiorelli@unicampania.it (A.F.); 2Department of Mental Health and Public Medicine, University of Campania “Luigi Vanvitelli”, 80131 Naples, Italy; 3Head and Neck Surgery Unit, University of Campania “Luigi Vanvitelli”, 80131 Naples, Italy; giovanni.motta@studenti.unicampania.it (G.M.); evaaurora.massimilla@unicampania.it (E.A.M.); gaetano.motta@unicampania.it (G.M.); 4Thoracic Surgery Unit, Sant’Andrea Hospital, Sapienza University, 00185 Rome, Italy; erinoangelo.rendina@uniroma1.it (E.A.R.); valentina.peritore@uniroma1.it (V.P.); claudio.andreetti@uniroma1.it (C.A.); 5Department of Precision Medicine, University of Campania “Luigi Vanvitelli”, 80131 Naples, Italy; antonello.sica@fastwebnet.it

**Keywords:** mediastinitis, descending necrotizing mediastinitis, thoracic surgery, otolaryngology

## Abstract

**Background:** Descending necrotizing mediastinitis (DNM) is a life-threatening condition, generally caused by downward dissemination of oropharyngeal infections through cervical fascial planes. Mediastinal drainage is conventionally achieved by thoracotomy, but a Video-Assisted Thoracoscopic Surgery (VATS) approach is gaining interest due to the reduced invasiveness of procedure. We aimed to evaluate the effectiveness of VATS treatment in patients with DNM. **Methods:** We conducted a retrospective multicenter study including patients with descending mediastinitis that underwent mediastinal drainage through VATS (VATS group) or thoracotomy (thoracotomy group), both in association with cervical drainage. Patients with mediastinitis secondary to cardiac, pulmonary, or esophageal surgery were excluded. The intergroup differences regarding surgical outcome and postoperative morbidity and mortality were compared. **Results:** A total of 21 patients were treated for descending mediastinitis during the study period. Cervicotomy and thoracotomy were performed in 15 patients (71%), while cervicotomy and VATS were performed in 6 patients (29%). There were no significant differences in surgical outcome, postoperative morbidity, and mortality between groups. VATS treatment was not associated with a higher complication rate. Patients in the VATS group had a shorter operative time (*p* = 0.016) and shorter ICU stay (*p* = 0.026). **Conclusions:** VATS treatment of DNM is safe and effective. The comparison with thoracotomy showed no significant differences in postoperative morbidity and mortality. The VATS approach is associated with a shorter operative time and ICU stay than thoracotomy.

## 1. Introduction

Descending necrotizing mediastinitis (DNM) is a rare but life-threatening condition, generally caused by the downward dissemination of oropharyngeal infections through the cervical fascial planes [1]. Soft-tissue abscesses of the neck usually arise from common pharyngitis, salivary gland infections, tonsillitis, odontogenic infections, cervical trauma, epiglottitis, or jugular intravenous drug use [2]. Odontogenic infections are the most common cause of DNM, as they have a series of potential routes of spread to nearby spaces, such as the submandibular, sublingual, parapharyngeal, masticator, and buccal. [3]. Risk factors are poor dental hygiene, drug and alcohol abuse, immunodeficiency syndromes, low socioeconomic status, diabetes, and cardiovascular diseases [4]. The clinical presentation of DNM is often non-specific, with fever, cough, cervical pain, and swelling being the most common presenting symptoms [5]. In addition, DNM is an extremely time-sensitive condition, with a reported mortality rate that ranges between 17.5 and 40% [6].

DNM diagnosis is based on clinical manifestations of severe infection, history of oropharyngeal infection, and suggestive radiological findings of mediastinal involvement [7]. Primary infection site control, surgical debridement, broad-spectrum antibiotic therapy, and early diagnosis are the key elements in mediastinitis management that unquestionably need to be addressed in a multidisciplinary setting [6,8].

The conventional surgical strategy for mediastinal drainage in patients with DNM is lateral/posterolateral thoracotomy [1,9], thanks to the optimal exposure of mediastinum, thoracic cavity, and pericardium that it guarantees. Mediastinal Video-Assisted Thoracoscopic Surgery (VATS) drainage was first proposed by Roberts et al. [10] in 1997, who highlighted the possibility of achieving sufficient drainage with the decreased morbidity of a minimally invasive approach. Also, subxyphoideal thoracoscopy is a possible surgical approach, though it has some limitations in posterior/lower mediastinum exploration [11].

The thoracoscopic approach to DNM is gaining interest due to its reduced invasiveness, particularly in elderly patients or patients requiring a bilateral approach [12]. At present, there is no consensus concerning the best approach. We aimed to evaluate the effectiveness of Video-Assisted Thoracoscopic Surgery treatment for patients suffering from DNM, comparing the surgical outcome and postoperative course of patients undergoing VATS versus thoracotomy for mediastinal drainage.

## 2. Materials and Methods

### 2.1. Study Design

This was a retrospective multicenter observational study conducted at the Thoracic Surgery Unit and the Head and Neck Surgery Unit of the University of Campania “Luigi Vanvitelli”, Naples, Italy, and the Thoracic Surgery Unit of Sant’Andrea University Hospital “La Sapienza”, Rome, Lazio, Italy. The Institutional Review Boards of the University of Campania “Luigi Vanvitelli” and of Sant’Andrea University Hospital “La Sapienza” waived the need for ethical approval due to the retrospective nature of the study since there was no modification to the standard of care for the patients. In all cases, informed written consent was obtained from the patients for the treatment and for the anonymous use of their data for scientific purposes. All procedures were performed in accordance with international guidelines; with the Helsinki Declaration of 1975, revised in 1983; and the rules of the Italian laws of privacy.

We retrospectively analyzed the data of consecutive patients undergoing surgical treatment for DNM from January 2013 to January 2023 at our institutions. All patients aged > 18 years with complete data and follow-up for at least 6 months postoperatively were included in the analysis. We excluded the data of patients (i) affected by mediastinitis secondary to cardiac, pulmonary, or esophageal surgery and (ii) with incomplete data or follow-up.

The patients were divided into two groups based on whether they underwent mediastinal drainage through thoracotomy (thoracotomy group) or VATS (VATS group). All the patients underwent concomitant cervical drainage through cervicotomy. The intergroup differences regarding surgical outcome and postoperative morbidity and mortality were statistically compared to assess the effectiveness of VATS treatment in patients with DNM.

### 2.2. Study Population

Anamnestic data, including age, gender comorbidities, time to diagnosis, time from diagnosis to surgical treatment, and mediastinitis clinical presentation, were recorded. Operative time (minutes), peri-operative complications, conversion rate, length of chest drainage (days), length of hospital stay (LHOS, days), length of intensive care unit (ICU) stay (days), reoperation rate, laboratory data, morbidity, and mortality were also recorded.

Diagnosis of DNM was based on clinical, radiological, and intraoperative findings according to Estrera’s criteria [13]: (1) clinical manifestations of severe infection; (2) demonstration of characteristic radiographic findings; (3) documentation of necrotizing mediastinal infection at operation; and (4) establishment of the relationship of oropharyngeal or cervical infection with the development of the necrotizing mediastinal process.

Preoperative radiographic images of two patients with DNM showing signs of neck and mediastinal involvement are shown in Figure 1 and Figure 2.

The extent of DNM was classified according to the criteria recently proposed by Sugio et al. [14]: type I—DNM limited to the area above the carina; and type II—DNM extending into the lower mediastinum, further divided into type IIA—anterior lower mediastinum; type IIB—anterior and posterior lower mediastinum; and type IIC—posterior lower mediastinum. Postoperative complications were classified according to the Systematic Classification of Morbidity and Mortality After Thoracic Surgery [15] in two categories: minor complications (grade I, requiring no treatment; grade II, requiring pharmacologic treatment or minor intervention only) and major complications (grade IIIa, requiring intervention without general anesthesia; grade IIIb, requiring intervention under general anesthesia; grade IVa, requiring intensive care unit management and life support with single organ dysfunction; grade IVb, requiring intensive care unit management and life support with multi-organ dysfunction).

### 2.3. Treatment and Surgical Procedure

All the patients underwent a conjoint surgical procedure of cervical and mediastinal drainage and debridement, performed by the Thoracic Surgery and Otolaryngology teams. Broad-spectrum antibiotics were administered until the specific pathogens responsible for the infections were identified.

The surgical procedures were carried out under general anesthesia and selective lung ventilation. Thoracotomy was performed at the fourth or fifth intercostal space. VATS was performed through a standard anterior triportal approach.

The surgical strategy was planned in relation to the primary source of infection and to a cervical and mediastinal level of involvement, with the goal to drain soft-tissue abscesses and to preserve airway functionality. When needed, a bilateral approach was carried out.

At the end of the procedure, two cervical closed-system suction drains (size 6–8 Fr) and two chest drainage tubes connected to an underwater seal chest drain system (size 28 Fr or 30 Fr) were placed. The two chest drainage tubes were placed in the mediastinum and in the pleural cavity. In case of bilateral involvement, additional cervical or thoracic drainage tubes were placed to ensure adequate drainage.

The chest drainage tubes were removed when the amount of fluid drained was less than 250 mL in 24 h, in the absence of air leaks and with evidence of negative pleural fluid culture.

After surgery, all the patients were admitted to the ICU for at least 12 h of intensive monitoring until their clinical conditions were stable, and then they were transferred to the surgical ward. A case of mediastinal drainage through thoracotomy is shown in Figure 3 and a case of VATS mediastinal drainage is represented in Figure 4.

### 2.4. Treatment Outcomes

The primary outcome of our study was 30-day mortality. Secondary outcomes were postoperative complications, operative time, reoperation rate, length of ICU stay, length of hospital stay, and chest tube duration.

### 2.5. Statistical Analysis

Data were expressed as mean ± standard deviation (SD) for continuous variables and as absolute number and percentage for categorical variables. The Shapiro–Wilk test was used to assess the normality of data. Non-normally distributed variables were expressed as median and interquartile range (IQR). Differences between groups were evaluated with Fisher’s exact test for categorical variables, Student’s *t*-test for continuous variables, and the Mann–Whitney U test for non-normally distributed continuous variables. The correlation between major complications and selected risk factors was investigated through univariable and multivariable analysis. A *p*-value less than 0.05 was considered statistically significant. MedCalc statistical software (Version 12.3, Broekstraat 52; Mariakerke, Belgium) was used for this analysis.

## 3. Results

During the study period, 23 patients underwent combined thoracic and otolaryngological surgical treatment for DNM. Two patients were excluded from the study due to incomplete follow-up. Therefore, our study population included 21 patients: 15 patients (71%) underwent cervicotomy and thoracotomy, while 6 patients (29%) underwent cervicotomy and VATS. Two patients in the VATS group (33%) underwent bilateral VATS. The two groups were homogeneous in terms of anamnestic data, comorbidities, and the mediastinal extent and source of infection (Table 1).

Intraoperative and postoperative outcome comparison is shown in Table 2.

The mean age of the population was 59 years, and the majority of patients were male (66%). The most common source of infection was odontogenic, and the most common isolated pathogen was Staphylococcus aureus (47%).

The median time to diagnosis of DNM was 1 day, while the median time from diagnosis to surgical treatment was 0.8 days, without significant differences between groups (*p* = 0.19 and *p* = 0.23, respectively).

The mean length of hospital stay was 42 days, which was a similar result between groups (*p* = 0.82). There were no significant differences regarding chest tube duration (*p* = 0.36), transfusion rate (*p* = 1), and reoperation rate (*p* = 1).

Patients that underwent VATS had a significantly shorter mean operative time (215 ± 20 vs. 250 ± 41 min, *p* = 0.016) and shorter ICU stay (12 ± 3.5 vs. 16 ± 4 days, *p* = 0.026).

The postoperative complication rate did not differ between groups; 10 patients had minor complications (47%) and 8 patients had major complications (38%).

Respiratory failure was the most common major complication (three patients, 14%), and in all three cases the patients received mechanical ventilation and temporary tracheostomy. One patient was subjected to percutaneous endoscopic gastrostomy (PEG) during the prolonged ICU stay. In the VATS group, none of the patients underwent conversion to thoracotomy during surgery.

There were no significant differences in 30-day and 90-day mortality rate between the two groups.

Univariable and multivariable logistic regression associated immunosuppression and age > 60 years with a higher incidence of postoperative major complications (Table 3).

## 4. Discussion

In our study, the comparison between patients that underwent VATS versus thoracotomy for DNM treatment showed no significant differences in terms of length of hospital stay, chest drainage duration, reoperation rate, and postoperative complications, and 30-day mortality, the primary outcome of our study, was comparable between groups, as with 90-day mortality. Interestingly, the patients in the VATS group had shorter operative time and shorter ICU stay. Univariable and multivariable analysis demonstrated that VATS treatment was not associated with a higher complication rate. Our results therefore suggest that VATS treatment of DNM is a safe and effective alternative to thoracotomy.

In the literature, the largest series of patients that underwent VATS treatment for DNM was analyzed by Tanaka et al. [16]. In their series of patients suffering from DNM, 83 patients underwent VATS, and 58 patients underwent thoracotomy. Patients with a poor performance status underwent VATS more frequently, while patients with type IIB mediastinitis generally underwent thoracotomy. The comparison between VATS and thoracotomy found no clinical and statistical differences in prognosis and surgical outcome. Patients in the VATS group had a higher rate of postoperative complication (53.0% vs. 24.1%) and reoperation (37.9% vs. 15.5%) than those who underwent thoracotomy, but the complications were not serious and could be easily addressed.

The patients in the thoracotomy group had a significantly longer operative time than those who underwent VATS. The results of this study are in concordance with ours regarding prognosis, surgical outcomes, and mortality of patients, without significant differences between groups. Also, the reduction in operative time in the VATS group is congruent with our results: a reduced general anesthesia duration is considered especially beneficial for elderly patients and patients with comorbidities [17], along with the advantage of smaller incisions and reduced postoperative pain associated with minimally invasive thoracoscopic procedures when compared with open surgery [18,19].

On a related note, Liang et al. [20] reported VATS treatment of 48 patients with acute necrotizing mediastinitis (ANM), comparing them with a group of 16 patients that underwent thoracotomy. Postoperative outcomes, including chest drainage duration, length of hospital stay, blood transfusion, reoperation rate, and mortality were comparable between groups. Even though we should remark that this study included patients with ANM in addition to DNM, its results are similar to ours in terms of surgical outcome and mortality rate. Other less recent studies on the matter, interesting to report as they represent the first published experiences of VATS mediastinal drainage for DNM, reported small series of patients with results similar to ours [21,22].

Undoubtedly, early diagnosis and treatment is a key element in DNM management. The reported mortality rate is highly variable: while some studies report a mortality between 16 and 20% [23,24], some others report a mortality of 30–40% [25,26]. It has been proved that early detection aided by timely diagnostic imaging is fundamental due to the rapid progression of DNM [27]. The mean time to treatment in the recent literature is about 1.5 days [3,14,28], and it is associated with a 30-day mortality lower than 20%. In our study, the median time to treatment was 0.8 days, and the 30-day mortality was 9%; hence, we may assume that the rapid treatment had a favorable influence on the survival rate.

Regarding prognostic factors, we observed that in our study, immunosuppression and age > 60 years were associated with a higher rate of postoperative major complications, in accordance with the literature [29,30]. VATS drainage was not associated with worse prognosis. The feasibility of VATS mediastinal drainage does not imply that all cases of DNM can be treated with VATS. If sufficient drainage of the mediastinum cannot be achieved through minimally invasive surgery, conversion to thoracotomy should be considered [16,31].

The preoperative study of chest CT guides the choice of the best approach for each patient, considering the possible limitations of VATS surgery in relation to the extent of the infection and structures involved [32].

Some limitations of our study are the retrospective nature and the relatively small sample size, which need to be considered in relation to the rarity of DNM. Also, the allocation to VATS or thoracotomy treatment could have been subjected to selection bias, since the choice of the approach was based on several considerations, including anatomical and functional features.

The outcomes of both VATS and thoracotomy mediastinal drainage are strictly dependent on the primary infection site control, comprehensive cervical drainage, antibiotic therapy, and all the procedures that are necessary to treat the incidental complications that may occur during the postoperative period. Cooperation between the thoracic, otolaryngology, maxillofacial, anesthesia, and infectious diseases departments allows an integrated and timely treatment of this threatening condition.

Future larger, prospective clinical trials are required to confirm our findings and guide physicians in the choice of the appropriate surgical technique for each case.

## 5. Conclusions

VATS treatment of descending necrotizing mediastinitis is safe and effective, while being comparable with thoracotomy in terms of surgical outcome, morbidity, and mortality. The VATS approach is associated with shorter operative time and ICU stay than thoracotomy. Early diagnosis and treatment are the key elements of DNM management.

## Figures and Tables

**Figure 1 jcm-13-02440-f001:**
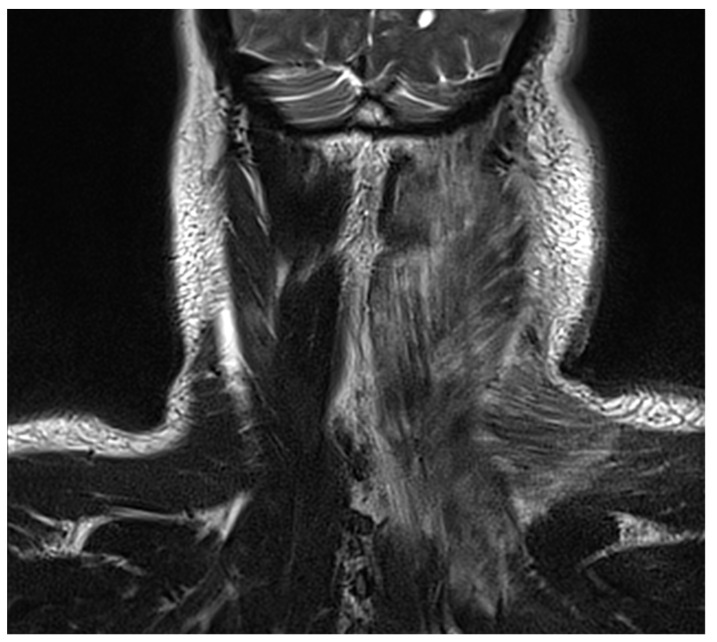
Cervical MRI image showing a neck abscess in a 44-year-old patient with DNM, treated with right cervicotomy and VATS mediastinal drainage.

**Figure 2 jcm-13-02440-f002:**
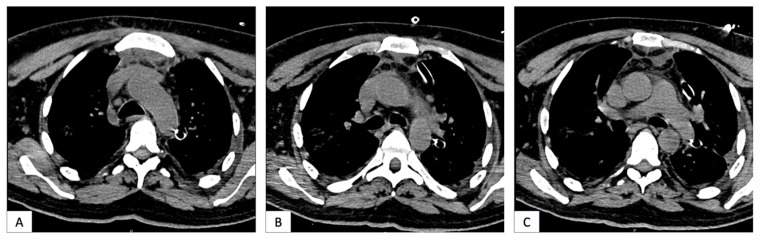
(**A**–**C**): Chest CT of a 43-year-old patient showing mediastinal involvement of DNM. The patient was treated with left cervicotomy and underwent a left thoracotomy for mediastinal drainage.

**Figure 3 jcm-13-02440-f003:**
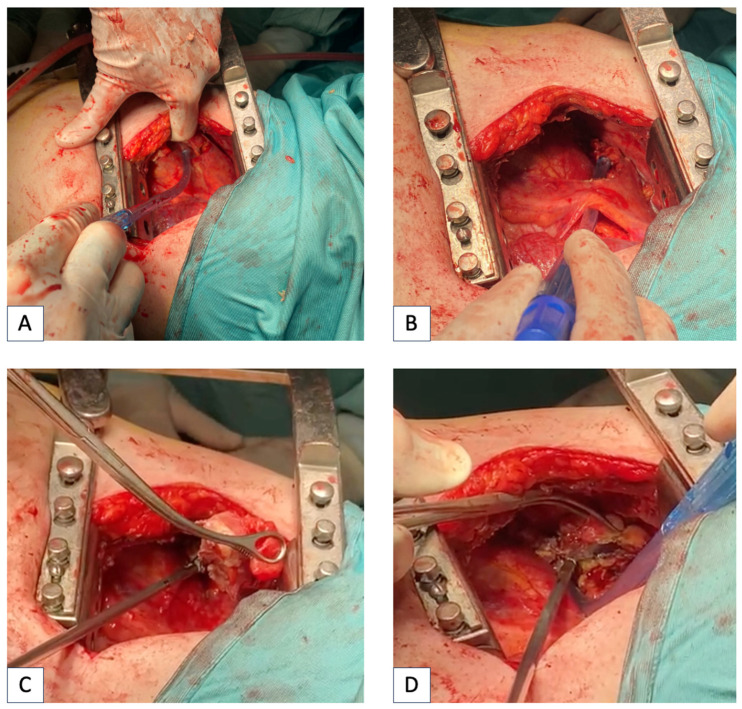
Mediastinal drainage through right thoracotomy in a patient suffering from DNM. (**A**): Exploration of mediastinum showing pericardial thickening and mediastinal abscess. (**B**): Anterior mediastinal exploration and debridement. (**C**,**D**): Dissection of necrotic and purulent tissue in the anterior mediastinum.

**Figure 4 jcm-13-02440-f004:**
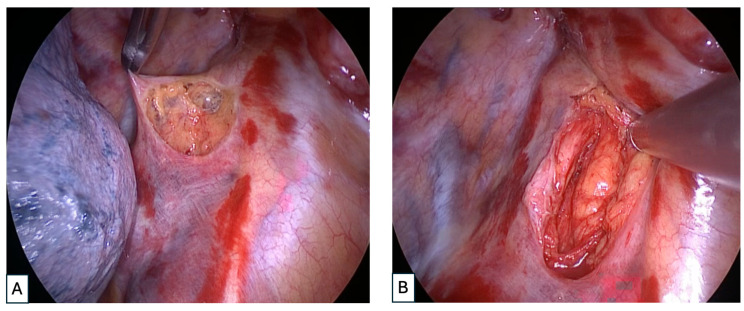
(**A**,**B**): Intraoperative images of anterior mediastinal dissection during right triportal VATS for DNM.

**Table 1 jcm-13-02440-t001:** Characteristics of study population.

Variables	Total (*n* = 21)	Thoracotomy Group (*n* = 15)	VATS Group (*n* = 6)	*p*-Value
Age (years), Mdn (IQR)	59 (51–67)	60 (54–67)	57 (50–65)	0.41
Gender (male), *n* (%)	14 (66)	11 (73)	3 (50)	0.35
BMI (Kg/m^2^), M ± SD	23.6 ± 3	23.8 ± 3	22.2 ± 3	0.21
Smoking, *n* (%)	12 (57)	9 (60)	3 (50)	1
Alcohol use, *n* (%)	9 (42)	7 (46)	2 (33)	0.65
Comorbidities, *n* (%):				
Diabetes	5 (23)	4 (26)	1 (16)	1
Hypertension	7 (33)	4 (26)	3 (50)	0.35
History of cancer	3 (14)	2 (13)	1 (16)	1
Immunosuppression	3 (14)	2 (13)	1 (16)	1
Source of infection *n* (%):				
Odontogenic	6 (28)	4 (26)	2 (33)	1
Oral	3 (14)	2 (13)	1 (16)	1
Pharyngeal	7 (33)	5 (33)	2 (16)	1
Cervical	5 (24)	4 (26)	1 (33)	1
Level of mediastinal extent *, *n* (%):				
Type I	9 (42)	6 (40)	3 (50)	0.63
Type IIA	2 (9)	1 (7)	1 (16)	0.5
Type IIB	6 (28)	5 (33)	1 (16)	0.62
Type IIC	4 (20)	3 (20)	1 (16)	1
Time to diagnosis (days), Mdn (IQR)	1 (1–1,75)	1 (1–2,75)	1 (1–1)	0.19
Time from diagnosis to surgical treatment, (days), Mdn (IQR)	0.8 (0.5–1.5)	0.8 (0.5–1.2)	1 (0.5–1.5)	0.23
Type of surgery, *n* (%):				
Thoracotomy + cervicotomy	/	15 (100)	/	/
VATS + cervicotomy	/	/	4 (66)	/
Bilateral VATS + cervicotomy	/	/	2 (33)	/
Pathogens isolated by culture, *n* (%):				
*Staphylococcus aureus*	10 (47)	8 (53)	2 (33)	0.63
*Streptococcus* spp.	4 (19)	2 (13)	2 (33)	0.54
*Enterobacteriaceae*	3 (14)	2 (13)	1 (16)	1
Other	4 (19)	3 (20)	1 (16)	1
Preoperative laboratory data,				
(M ± SD):				
White blood cells (/uL)	14.010 ± 1340	14.080 ± 1050	13.950 ± 1173	0.42
Hemoglobin (g/dL)	12.3 ± 1.4	12.4 ± 1.7	12.0 ± 1.4	0.37
Platelet count (×10^3^/uL)	220 ± 89	225 ± 78	215 ± 60	0.5
Total protein (g/dL)	6.9 ± 1.8	7.2 ± 1.9	6.8 ± 2.1	0.48

* According to Sugio et al.’s [14,16] classification. Mdn: median. IQR; interquartile range. M: mean. SD: standard deviation. VATS: video-assisted thoracoscopic surgery. BMI: body mass index.

**Table 2 jcm-13-02440-t002:** VATS vs. thoracotomy surgical outcome comparison.

Variables	Total (*n* = 21)	Thoracotomy Group (*n* = 15)	VATS Group (*n* = 6)	*p*-Value
Operative time (minutes), M ± SD	238 ± 38	250 ± 41	215 ± 20	0.016
Chest tube duration, Mdn (IQR)	15 (10–18)	16 (10–20)	13 (9–17)	0.36
Blood transfusion, *n* (%)	7 (33)	5 (33)	2 (33)	0.65
Reoperation, *n* (%)	3 (14)	2 (13)	1 (16)	1
ICU stay (days), M ± SD	15 ± 4	16 ± 4	12 ± 3.5	0.026
LHOS (days),	42 ± 10	43 ± 8	42 ± 9	0.82
Conversion, *n* (%)	0	/	0	0
Minor complications (grade I-II *)	10 (47)	7 (46)	3 (50)	0.35
Major complications (grade III-IV *):	8 (38)	6 (40)	2 (33)	1
Respiratory failure	3 (14)	2 (13)	1 (16)	/
Septic shock	2 (10)	2 (13)	0	/
Renal failure	1 (4)	1 (6)	0	/
Arrhythmias	2 (10)	1 (6)	1 (16)	/
30-day mortality, *n* (%)	2 (9)	1 (6)	1 (16)	1
90-day mortality, *n* (%)	3 (14)	2 (13)	1 (16)	1

M: mean. SD: standard deviation. LHOS: length of hospital stay. ICU: intensive care unit. VATS: video-assisted thoracoscopic surgery. * According to the Systematic Classification of Morbidity and Mortality After Thoracic Surgery (Seely AJE et al. [14,16]).

**Table 3 jcm-13-02440-t003:** Univariable and multivariable analysis of factors associated with major complications (dependent variable).

Variables	Univariable		Multivariable	
	Odds Ratio	*p*-Value	Odds Ratio	*p*-Value
Age (>60 years)	1.34 (CI: 0.59–3.08)	0.04	2.15 (CI 1.04–4.76)	0.001
Gender	3.20 (CI: 1.48–5.42)	0.23	-	-
Smoking	0.55 (CI: 0.13–1.93)	0.48	-	-
Immunosuppression	2.15 (CI: 0.98–6.17)	0.003	3.02 (CI 1.56–5.12)	0.002
Diabetes	1.93 (CI: 1.03–5.41)	0.49		
History of cancer	1.82 (CI: 0.67–3.85)	0.62	-	-
Reoperation	1.29 (CI 0.33–4.03)	0.15	-	-
Thoracotomy	0.84 (CI 0.24–2.76)	0.34	-	-
VATS	1.36 (CI: 0.46–2.40)	0.69	-	-

CI: confidence interval. VATS: video-assisted thoracoscopic surgery.

## Data Availability

The data presented in this study are available on request from the corresponding author.

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
