# Peer review of "Multidisciplinary Management of Descending Necrotizing Mediastinitis: Is Thoracoscopic Treatment Feasible?"

_jcm, 2024, doi:10.3390/jcm13082440_

Round 1
Reviewer 1 Report
Comments and Suggestions for Authors
The authors evaluate the effectiveness of videothoracoscopy (VATS) in treatment in patients with Descending Necrotizing Mediastinitis (DNM). They concluded that the treatment of descending necrotizing mediastinitis requires accurate preoperative planning and multidisciplinary approach. Minimally invasive VATS approach is feasible, but if sufficient drainage cannot be achieved, thoracotomy should be performed.
I read the study with great interest. My comments and suggestions for improvement are as follows:
1. First of all, the number of 20 authors is extremely high for a simple observational study. As can be seen from the authors' contributions, some authors have made a very limited contribution. This number should be significantly reduced. The authors who only treated the patients did not deserve authorship, it is their everyday job. Only authors with a significant contribution should be listed as authors.
2. The authors have used several abbreviations in their abstract (VATS, ICU, DNM). Please expand the abbreviation the first time mentioned in the abstract. The same follows in introduction (expand VATS when the first time mentioned).
3. The introduction is very short and not informative. It should be more detailed in terms of causes, diagnostic modalities and treatment options than presented.
4. The authors state that the study was approved by the local ethics committee of the University of Campania “Luigi Vanvitelli”, the coordinating centre of the study, and subsequently by the local ethics committees of each participating centre. If the study has been approved, please provide the approval number and date of approval for each centre involved in the study. It is inconsistent to state that the study has been approved and not provide a specific approval number in the next paragraph!
5. Chapters 2.3., 2.4. and 2.6. (Figures) should be deleted. All figures should be included in the chapter in which they are mentioned. In addition, the next chapter should be renumbered.
6. Please specify the type and diameter of drainage used for the procedure.
7. Please add a new chapter in the methodology – treatment outcomes and describe the primary and secondary outcomes of the study.
8. Which statistical test was used to check the normality of the data distribution? Since it is clear from the methodology that all variables are not normally distributed, is it hard to believe that some of the variables do not have a normal distribution? Please clarify this!
9. In addition to the previous comment, only normally distributed variables should be reported as mean (SD). Non-normally distributed variables should be expressed as median (IQR) and tested with an appropriate statistical test (e.g. Mann-Whitney U test).
10. Tables – All abbreviations mentioned in the table should be listed in the table legend. E.g. VATS, M, SD, BMI, etc.
11. Tables should not be listed as separate chapters (as with figures). Please revise.
12. The discussion needs improvement. It consists largely of repeating known facts from the medical literature that are not up for discussion. There is no need to repeat general data from the literature (this should be moved to the introduction). Authors should focus on the results of the main objectives of the study and write the discussion in a few consecutive paragraphs (without headings): The first paragraph should summarise the main findings of the study (without repeating the data); Next, the authors should provide coherent comparison of their results with the existing literature, focusing on the main objectives; Finally, limitations of the study and implications for practice and perspective for further studies.
13. Conclusions are general and should be more focused on main findings of the study.
14. The bibliography is incomplete. The authors should carry out a better literature search. More recent references should be cited. At least 30 recent references are required for an original article.
15. The quality of the English language should be improved. There are many expressions, terms and sentences that need significant amount of revisions. The text would benefit from professional editing.
Comments on the Quality of English LanguageThe quality of the English language should be improved. There are many expressions, terms and sentences that need significant amount of revisions. The text would benefit from professional editing.
Author Response
To the Editor in Chief of Journal of Clinical Medicine
We re-submit our article title: “Multidisciplinary Management of Descending Necrotizing Mediastinitis: is Thoracoscopic Treatment Feasible?”, manuscript ID: jcm-2939807.
Type of manuscript: Article
Authors: Beatrice Leonardi, Giovanni Natale, Caterina Sagnelli *, Antonio Marella, Francesco Leone, Francesca Capasso, Noemi Maria Giorgiano, Davide Gerardo Pica, Rosa Mirra, Vincenzo Di Filippo, Gaetana Messina, Giovanni Vicidomini, Giovanni Motta, Eva Aurora Massimilla, Gaetano Motta, Erino Angelo Rendina, Valentina Peritone, Claudio Andreetti, Alfonso Fiorelli, Antonello Sica.
The following changes (shown underlined). The manuscript has been improved according to the suggestions of the reviewer: I thank the Editor for the opportunity of reviewing this paper.
Reviewer(s)' Comments to Author:
Reviewer #1: The authors evaluate the effectiveness of videothoracoscopy (VATS) in treatment in patients with Descending Necrotizing Mediastinitis (DNM). They concluded that the treatment of descending necrotizing mediastinitis requires accurate preoperative planning and multidisciplinary approach. Minimally invasive VATS approach is feasible, but if sufficient drainage cannot be achieved, thoracotomy should be performed.
I read the study with great interest. My comments and suggestions for improvement are as follows:
Point 1: First of all, the number of 20 authors is extremely high for a simple observational study. As can be seen from the authors' contributions, some authors have made a very limited contribution. This number should be significantly reduced. The authors who only treated the patients did not deserve authorship, it is their everyday job. Only authors with a significant contribution should be listed as authors.
Answer to the Reviewer point 1: The author numbers is indeed high, but this does not mean that some of them have not contributed actively. This work is observational but multicentric, therefore data retrieving and follow up of the patient for scientific purposes, in addition to validation of the study findings, required the collaboration of all the included authors. None of the authors simply treated the patients, but all of them contributed to the research project. We think that reducing the number of authors after their work for this research project would be unfair.
Point 2: The authors have used several abbreviations in their abstract (VATS, ICU, DNM). Please expand the abbreviation the first time mentioned in the abstract. The same follows in introduction (expand VATS when the first time mentioned).
Answer to the Reviewer point 2: The manuscript has been improved according to the suggestions of the reviewer: We expanded the abbreviations when first mentioned in the abstract and introduction, as suggested.
“Descending necrotizing mediastinitis (DNM)” (Changes in line 28, page 1)
“Video-Assisted Thoracoscopic Surgery (VATS)” (Changes in lines 30-31, page 1)
“(…) shorter intensive care unit stay (p = 0.026)” (Changes in line 44, page 1)
“Video-Assisted Thoracoscopic Surgery (VATS) treatment for patients suffering (…)” (Changes in line 79, page 2)
Point 3: The introduction is very short and not informative. It should be more detailed in terms of causes, diagnostic modalities and treatment options than presented.
Answer to the Reviewer point 3: The manuscript has been improved according to the suggestions of the reviewer. We revised the introduction, adding context information on DNM causes, diagnosis and treatment options. (Changes in lines 52-81, page 2)
Point 4: The authors state that the study was approved by the local ethics committee of the University of Campania “Luigi Vanvitelli”, the coordinating centre of the study, and subsequently by the local ethics committees of each participating centre. If the study has been approved, please provide the approval number and date of approval for each centre involved in the study. It is inconsistent to state that the study has been approved and not provide a specific approval number in the next paragraph!
Answer to the Reviewer point 4: We thank the Reviewer for reviewing our manuscript and for commenting. We revised the ethics statement to clarify that ethic approval was waived by the local IRB due to the retrospective nature of the study.
“The Institutional Review Board of of University of Campania “Luigi Vanvitelli” and of Sant'Andrea Hospital University “La Sapienza” waived the need for ethical approval due to the retrospective nature of the study since there was no modification to the standard of care for the patients. In all cases, an informed written consent was obtained from the patients for the treatment and for the anonymous use of their data for scientific purposes. All the procedures were performed in accordance with the international guidelines, with the Helsinki Declaration of 1975, revised in 1983 and the rules of the Italian laws of privacy.” (Changes in lines 89-94, page 2)
Point 5: Chapters 2.3., 2.4. and 2.6. (Figures) should be deleted. All figures should be included in the chapter in which they are mentioned. In addition, the next chapter should be renumbered.
Answer to the Reviewer point 5: We revised the figures denomination and placement, removing the corresponding chapters and renumbering the chapters of Materials and Methods section accordingly. (Changes in pages 3, 4, 5).
Point 6: Please specify the type and diameter of drainage used for the procedure.
Answer to the Reviewer point 6: We specified the type and diameter of drainage used for the procedure.
“At the end of the procedure, two cervical closed-system suction drains (size 6-8 Fr) and two chest drainage tubes connected to an underwater seal chest drain system (size 28Fr or 30Fr) were placed. The two chest drainages were placed in the mediastinum and in the pleural cavity. In case of bilateral involvement, additional cervical or thoracic drainages were placed to ensure adequate drainage.” (Changes in lines 158-162, page 4)
Point 7: Please add a new chapter in the methodology – treatment outcomes and describe the primary and secondary outcomes of the study.
Answer to the Reviewer point 7: The manuscript has been improved according to the suggestions of the reviewer. We added the new methodology section “treatment outcomes” as suggested.
“2.4 Treatment Outcomes
The primary outcome of our study was 30-day mortality. Secondary outcomes were postoperative complications, operative time, reoperation rate, length of ICU stay, length of hospital stay and chest tube duration. “ (Changes in lines 173-176, page 9)
Point 8: Which statistical test was used to check the normality of the data distribution? Since it is clear from the methodology that all variables are not normally distributed, is it hard to believe that some of the variables do not have a normal distribution? Please clarify this!
Answer to the Reviewer point 8: Shapiro-Wilk test was performed to assess normality of data. We revised the text accordingly.
“Shapiro-Wilk test was used to assess normality of data.” (Changes in lines 179-180, page 5)
Point 9: In addition to the previous comment, only normally distributed variables should be reported as mean (SD). Non-normally distributed variables should be expressed as median (IQR) and tested with an appropriate statistical test (e.g. Mann-Whitney U test).
Answer to the Reviewer point 9: The observation of the reviewer has been accepted and the new manuscript has been modified accordingly. Non normally distributed variables were expressed as median and interquartile range (IQR) and tested with Mann-Whitney U test. We revised the tables accordingly.
“Non normally distributed variables were expressed as median and interquartile range (IQR). Differences between groups were evaluated with Fisher’s exact test for categorical variables, Student´s T-test for continuous variables and Mann-Whitney U test for not normally distributed continuous variables”. (Changes in lines 180-183, page 5)
Point 10: Tables – All abbreviations mentioned in the table should be listed in the table legend. E.g. VATS, M, SD, BMI, etc.
Answer to the Reviewer point 10: The observation of the reviewer has been accepted and the new manuscript has been modified accordingly. We revised the table legends and listed all the mentioned abbreviations. (Changes in pages 6, 7, 8).
Point 11: Tables should not be listed as separate chapters (as with figures). Please revise.
Answer to the Reviewer point 11: The manuscript has been improved according to the suggestions of the reviewer. We revised the manuscript and removed the table listing as separate chapters. (Changes in pages 6, 7, 8).
Point 12: The discussion needs improvement. It consists largely of repeating known facts from the medical literature that are not up for discussion. There is no need to repeat general data from the literature (this should be moved to the introduction). Authors should focus on the results of the main objectives of the study and write the discussion in a few consecutive paragraphs (without headings): The first paragraph should summarise the main findings of the study (without repeating the data); Next, the authors should provide coherent comparison of their results with the existing literature, focusing on the main objectives; Finally, limitations of the study and implications for practice and perspective for further studies.
Answer to the Reviewer point 12: The manuscript has been improved according to the suggestions of the reviewer. We revised the whole discussion as you suggested. (Changes in lines 236-299, pages 8,9)
Point 13: Conclusions are general and should be more focused on main findings of the study.
Answer to the Reviewer point 13: The observation of the reviewer has been accepted and the new manuscript has been modified accordingly. We revised our conclusions in the abstract and manuscript as follows:
“VATS treatment of DNM is safe and effective. The comparison with thoracotomy showed no significant differences in terms of postoperative morbidity and mortality. VATS approach is associated with shorter operative time and ICU stay than thoracotomy. “(Changes in lines 45-47, page 1)
“VATS treatment of descending necrotizing mediastinitis is safe and effective, while being comparable with thoracotomy in terms of surgical outcome, morbidity, and mortality. VATS approach is associated with shorter operative time and ICU stay than thoracotomy. Early diagnosis and treatment are the key elements of DNM management. (Changes in lines 301-304, page 10)
Point 14: The bibliography is incomplete. The authors should carry out a better literature search. More recent references should be cited. At least 30 recent references are required for an original article.
Answer to the Reviewer point 14: The observation of the reviewer has been accepted and the new manuscript has been modified accordingly. We implemented the bibliography through a new literature search.
Point 15: The quality of the English language should be improved. There are many expressions, terms and sentences that need significant amount of revisions. The text would benefit from professional editing.
Answer to the Reviewer point 15: The observation of the reviewer has been accepted and the new manuscript has been modified accordingly. We revised English language of the manuscript.
Reviewer #2:
Descending necrotizing mediastinitis is one of the critical clinical diseases with a high mortality rate. The authors retrospectively analyzed treatment cases from multiple centers and compared theoutcomes of minimally invasive VATS surgery and conventional open thoracotomy surgery. They concluded that The treatment of descending necrotizing mediastinitis requires accurate pre-operative planning and multidisciplinary approach. Minimally invasive VATS approach is feasible, but if sufficient drainage cannot be achieved, thoracotomy should be performed.
This is an interesting paper, as such readable cases often come from case reports. In the current medical environment, it is difficult to encounter a large number of such diseases. For thoracic surgeons, obtaining such a conclusion is obvious. If minimally invasive surgery can solve the problem, open surgery will definitely not be chosen.
Here are still some questions that the authors need to answer or to discuss
Point 1: In Figure 2, the patient underwent left cervical incision and right thoracic surgery, but the thoracic drainage tube was clearly placed in the patient's left chest, which is unreasonable.
Answer to the Reviewer point 1: The observation of the reviewer has been accepted and the new manuscript has been modified accordingly. We made a mistake in Figure 2 legend, we revised that. The thoracic surgery was performed on the left.
“Chest CT of a 43-year-old patient showing mediastinal involvement by DNM. The patient was treated with left cervicotomy and underwent a left thoracotomy for mediastinal drainage.” (Changes in lines 132-134, page 4)
Point 2: The most important treatment for DNM lies in timely detection and early operation and drainage. Have the authors considered the relationship between the time from diagnosis to surgery and treatment outcomes?
Answer to the Reviewer point 2: Yes, we evaluated time to diagnosis and time to treatment that resulted to be very short and without significant difference between groups. We analyzed time to treatment relation with outcome also in the discussion. We thank the Reviewer for reviewing our manuscript and for commenting. The manuscript has been improved according to the suggestions of the reviewer.
“Anamnestic data, including, age, gender comorbidities, time to diagnosis, time from diagnosis to surgical treatment and mediastinitis clinical presentation were recorded.” (Changes in lines 111-112, page 3)
“The mean age of the population was 59 years and the majority of patients were male (66%). The most common source of infection was odontogenic and the most common isolated pathogen was Staphylococcus aureus (47%). The median time to diagnosis of DNM was 1 day while the median time from diagnosis to surgical treatment was 0.8 days, without significant differences between groups (p = 0.19 and p = 0.23, respectively).” (Changes in lines 209-213, page 7)
“Undoubtedly, early diagnosis and treatment is a key element in DNM management. The reported mortality rate is highly variable: while some studies report a mortality between 16 and 20% [22, 23], some others report a mortality of 30-40% [24,25]. It has been proved that early detection aided by timely diagnostic imaging is fundamental due to the rapid progression of DNM [26]. The mean time to treatment in recent literature is about 1,5 days [3, 14, 27], and it is associated with a 30- day mortality lower than 20%. In our study the median time to treatment was 0.8 days, and the 30-day mortality was 9%, hence we may assume that the rapid treatment had a favorable influence on the survival rate.” (Changes in lines 271-278, page 9)
Point 3: The surgical areas involved in mediastinal infections are not consistent, and,relatively, the choice of surgical approach may also vary greatly. Have the authors considered the feasibility of performing surgery through the subxiphoid approach in cases of bilateral VATS or cases which infection dominated in anterior mediastinum?
Answer to the Reviewer point 3: The observation of the reviewer has been accepted and the new manuscript has been modified accordingly. We have considered the use of subxiphoid approach, but in our institutions, we prefer the exposure of triportal VATS for bilateral involvement in order to explore the entire mediastinum with ease.
“The conventional surgical strategy for mediastinal drainage in patients with DNM is lateral/posterolateral thoracotomy [1,9], thanks to the optimal exposure of mediastinum, thoracic cavity and pericardium that it guarantees. Mediastinal VATS drainage has been first proposed by Roberts et al. [10] in 1997, that highlighted the possibility to achieve sufficient drainage with the decreased morbidity of a minimally invasive approach. Also, subxyphoideal thoracoscopy is a possible surgical approach, though it has some limita-tions in the posterior/lower mediastinum exploration [11]” (Changes in lines 69-75, page 2)
Point 4: 4. The authors performed simultaneous neck and chest surgeries for these cases. Is simple neck surgery feasible for type I cases in the real world? Are there any patients who undergo chest surgery because neck surgery is not enough to cure them?
Answer to the Reviewer point 4: Yes, simple neck surgery is feasible for some cases of type I DNM, bearing in mind that the infection can progress. The patient could need a reintervention on the thorax if the infection is not well controlled. Surely, patients with compromised clinical condition benefit from aggressive surgical treatment including thorax exploration and drainage.
Reviewer #3: This well written paper may be improved by:
Point 1: Adding în text and also în conclusions the rapid management of the primary site which caused the mediastinitis, besides the actual mediastinitis treatment;
Answer to the Reviewer point 1: The manuscript has been improved according to the suggestions of the reviewer. We revised the text and diagnosis clarifying the topic of early diagnosis and treatment of the primary site of infection.
“Primary infection site control, surgical debridement and broad-spectrum antibiotic therapy and early diagnosis are the key elements in mediastinitis management, that unquestionably needs to be addressed in a multidisciplinary setting [6, 8].” (Changes in lines 66-68, page 2)
“The surgical strategy was planned in relation to the primary source of infection and to cervical and mediastinal level of involvement, with the goal to drain soft tissues abscesses and to preserve the airway functionality. When needed, a bilateral approach was carried out.” (Changes in lines 154-157, page 4)
VATS treatment of descending necrotizing mediastinitis is safe and effective, while being comparable with thoracotomy in terms of surgical outcome, morbidity and mor-tality. VATS approach is associated with shorter operative time and ICU stay than thoracotomy. Early diagnosis and treatment are the key elements of DNM management. (Changes in lines 302-305, page 10)
Point 2: Adding some intraoperative images from toracoscopic mediastinal drainage – would have a hudge impact on readers;
Answer to the Reviewer point 2: The observation of the reviewer has been accepted and the new manuscript has been modified accordingly.
Point 3: Number and quality of the reference must be improved.
With these adds, I am convinced that the paper is ready to be published, in my professional opinion.
Congratulations for the authors!
Answer to the Reviewer point 3: We thank the Reviewer for reviewing our manuscript and for commenting. We have updated the bibliography with a new bibilography search.
We thank the Editor and the Reviewers for helping us to improve our paper.
The manuscript has been read and approved by all the authors.
We also declare that we have no conflict of interest in connection with this paper. We sincerely hope that the enclosed manuscript can be accepted for publication in the Journal of Clinical Medicine
Prof. C. Sagnelli
Prof. A. Fiorelli
Dott.ssa A. Sica

Reviewer 2 Report
Comments and Suggestions for Authors
Descending necrotizing mediastinitis is one of the critical clinical diseases with a high mortality rate. The authors retrospectively analyzed treatment cases from multiple centers and compared theoutcomes of minimally invasive VATS surgery and conventional open thoracotomy surgery. They concluded that The treatment of descending necrotizing mediastinitis requires accurate pre-operative planning and multidisciplinary approach. Minimally invasive VATS approach is feasible, but if sufficient drainage cannot be achieved, thoracotomy should be performed.
This is an interesting paper, as such readable cases often come from case reports. In the current medical environment, it is difficult to encounter a large number of such diseases. For thoracic surgeons, obtaining such a conclusion is obvious. If minimally invasive surgery can solve the problem, open surgery will definitely not be chosen.
Here are still some questions that the authors need to answer or to discuss
1. In Figure 2, the patient underwent left cervical incision and right thoracic surgery, but the thoracic drainage tube was clearly placed in the patient's left chest, which is unreasonable.
2. The most important treatment for DNM lies in timely detection and early operation and drainage. Have the authors considered the relationship between the time from diagnosis to surgery and treatment outcomes?
3. The surgical areas involved in mediastinal infections are not consistent, and,relatively, the choice of surgical approach may also vary greatly. Have the authors considered the feasibility of performing surgery through the subxiphoid approach in cases of bilateral VATS or cases which infection dominated in anterior mediastinum?
4. The authors performed simultaneous neck and chest surgeries for these cases. Is simple neck surgery feasible for type I cases in the real world? Are there any patients who undergo chest surgery because neck surgery is not enough to cure them?
Comments on the Quality of English Languagenothing
Author Response
To the Editor in Chief of Journal of Clinical Medicine
We re-submit our article title: “Multidisciplinary Management of Descending Necrotizing Mediastinitis: is Thoracoscopic Treatment Feasible?”, manuscript ID: jcm-2939807.
Type of manuscript: Article
Authors: Beatrice Leonardi, Giovanni Natale, Caterina Sagnelli *, Antonio Marella, Francesco Leone, Francesca Capasso, Noemi Maria Giorgiano, Davide Gerardo Pica, Rosa Mirra, Vincenzo Di Filippo, Gaetana Messina, Giovanni Vicidomini, Giovanni Motta, Eva Aurora Massimilla, Gaetano Motta, Erino Angelo Rendina, Valentina Peritone, Claudio Andreetti, Alfonso Fiorelli, Antonello Sica.
The following changes (shown underlined). The manuscript has been improved according to the suggestions of the reviewer: I thank the Editor for the opportunity of reviewing this paper.
Reviewer(s)' Comments to Author:
Reviewer #1: The authors evaluate the effectiveness of videothoracoscopy (VATS) in treatment in patients with Descending Necrotizing Mediastinitis (DNM). They concluded that the treatment of descending necrotizing mediastinitis requires accurate preoperative planning and multidisciplinary approach. Minimally invasive VATS approach is feasible, but if sufficient drainage cannot be achieved, thoracotomy should be performed.
I read the study with great interest. My comments and suggestions for improvement are as follows:
Point 1: First of all, the number of 20 authors is extremely high for a simple observational study. As can be seen from the authors' contributions, some authors have made a very limited contribution. This number should be significantly reduced. The authors who only treated the patients did not deserve authorship, it is their everyday job. Only authors with a significant contribution should be listed as authors.
Answer to the Reviewer point 1: The author numbers is indeed high, but this does not mean that some of them have not contributed actively. This work is observational but multicentric, therefore data retrieving and follow up of the patient for scientific purposes, in addition to validation of the study findings, required the collaboration of all the included authors. None of the authors simply treated the patients, but all of them contributed to the research project. We think that reducing the number of authors after their work for this research project would be unfair.
Point 2: The authors have used several abbreviations in their abstract (VATS, ICU, DNM). Please expand the abbreviation the first time mentioned in the abstract. The same follows in introduction (expand VATS when the first time mentioned).
Answer to the Reviewer point 2: The manuscript has been improved according to the suggestions of the reviewer: We expanded the abbreviations when first mentioned in the abstract and introduction, as suggested.
“Descending necrotizing mediastinitis (DNM)” (Changes in line 28, page 1)
“Video-Assisted Thoracoscopic Surgery (VATS)” (Changes in lines 30-31, page 1)
“(…) shorter intensive care unit stay (p = 0.026)” (Changes in line 44, page 1)
“Video-Assisted Thoracoscopic Surgery (VATS) treatment for patients suffering (…)” (Changes in line 79, page 2)
Point 3: The introduction is very short and not informative. It should be more detailed in terms of causes, diagnostic modalities and treatment options than presented.
Answer to the Reviewer point 3: The manuscript has been improved according to the suggestions of the reviewer. We revised the introduction, adding context information on DNM causes, diagnosis and treatment options. (Changes in lines 52-81, page 2)
Point 4: The authors state that the study was approved by the local ethics committee of the University of Campania “Luigi Vanvitelli”, the coordinating centre of the study, and subsequently by the local ethics committees of each participating centre. If the study has been approved, please provide the approval number and date of approval for each centre involved in the study. It is inconsistent to state that the study has been approved and not provide a specific approval number in the next paragraph!
Answer to the Reviewer point 4: We thank the Reviewer for reviewing our manuscript and for commenting. We revised the ethics statement to clarify that ethic approval was waived by the local IRB due to the retrospective nature of the study.
“The Institutional Review Board of of University of Campania “Luigi Vanvitelli” and of Sant'Andrea Hospital University “La Sapienza” waived the need for ethical approval due to the retrospective nature of the study since there was no modification to the standard of care for the patients. In all cases, an informed written consent was obtained from the patients for the treatment and for the anonymous use of their data for scientific purposes. All the procedures were performed in accordance with the international guidelines, with the Helsinki Declaration of 1975, revised in 1983 and the rules of the Italian laws of privacy.” (Changes in lines 89-94, page 2)
Point 5: Chapters 2.3., 2.4. and 2.6. (Figures) should be deleted. All figures should be included in the chapter in which they are mentioned. In addition, the next chapter should be renumbered.
Answer to the Reviewer point 5: We revised the figures denomination and placement, removing the corresponding chapters and renumbering the chapters of Materials and Methods section accordingly. (Changes in pages 3, 4, 5).
Point 6: Please specify the type and diameter of drainage used for the procedure.
Answer to the Reviewer point 6: We specified the type and diameter of drainage used for the procedure.
“At the end of the procedure, two cervical closed-system suction drains (size 6-8 Fr) and two chest drainage tubes connected to an underwater seal chest drain system (size 28Fr or 30Fr) were placed. The two chest drainages were placed in the mediastinum and in the pleural cavity. In case of bilateral involvement, additional cervical or thoracic drainages were placed to ensure adequate drainage.” (Changes in lines 158-162, page 4)
Point 7: Please add a new chapter in the methodology – treatment outcomes and describe the primary and secondary outcomes of the study.
Answer to the Reviewer point 7: The manuscript has been improved according to the suggestions of the reviewer. We added the new methodology section “treatment outcomes” as suggested.
“2.4 Treatment Outcomes
The primary outcome of our study was 30-day mortality. Secondary outcomes were postoperative complications, operative time, reoperation rate, length of ICU stay, length of hospital stay and chest tube duration. “ (Changes in lines 173-176, page 9)
Point 8: Which statistical test was used to check the normality of the data distribution? Since it is clear from the methodology that all variables are not normally distributed, is it hard to believe that some of the variables do not have a normal distribution? Please clarify this!
Answer to the Reviewer point 8: Shapiro-Wilk test was performed to assess normality of data. We revised the text accordingly.
“Shapiro-Wilk test was used to assess normality of data.” (Changes in lines 179-180, page 5)
Point 9: In addition to the previous comment, only normally distributed variables should be reported as mean (SD). Non-normally distributed variables should be expressed as median (IQR) and tested with an appropriate statistical test (e.g. Mann-Whitney U test).
Answer to the Reviewer point 9: The observation of the reviewer has been accepted and the new manuscript has been modified accordingly. Non normally distributed variables were expressed as median and interquartile range (IQR) and tested with Mann-Whitney U test. We revised the tables accordingly.
“Non normally distributed variables were expressed as median and interquartile range (IQR). Differences between groups were evaluated with Fisher’s exact test for categorical variables, Student´s T-test for continuous variables and Mann-Whitney U test for not normally distributed continuous variables”. (Changes in lines 180-183, page 5)
Point 10: Tables – All abbreviations mentioned in the table should be listed in the table legend. E.g. VATS, M, SD, BMI, etc.
Answer to the Reviewer point 10: The observation of the reviewer has been accepted and the new manuscript has been modified accordingly. We revised the table legends and listed all the mentioned abbreviations. (Changes in pages 6, 7, 8).
Point 11: Tables should not be listed as separate chapters (as with figures). Please revise.
Answer to the Reviewer point 11: The manuscript has been improved according to the suggestions of the reviewer. We revised the manuscript and removed the table listing as separate chapters. (Changes in pages 6, 7, 8).
Point 12: The discussion needs improvement. It consists largely of repeating known facts from the medical literature that are not up for discussion. There is no need to repeat general data from the literature (this should be moved to the introduction). Authors should focus on the results of the main objectives of the study and write the discussion in a few consecutive paragraphs (without headings): The first paragraph should summarise the main findings of the study (without repeating the data); Next, the authors should provide coherent comparison of their results with the existing literature, focusing on the main objectives; Finally, limitations of the study and implications for practice and perspective for further studies.
Answer to the Reviewer point 12: The manuscript has been improved according to the suggestions of the reviewer. We revised the whole discussion as you suggested. (Changes in lines 236-299, pages 8,9)
Point 13: Conclusions are general and should be more focused on main findings of the study.
Answer to the Reviewer point 13: The observation of the reviewer has been accepted and the new manuscript has been modified accordingly. We revised our conclusions in the abstract and manuscript as follows:
“VATS treatment of DNM is safe and effective. The comparison with thoracotomy showed no significant differences in terms of postoperative morbidity and mortality. VATS approach is associated with shorter operative time and ICU stay than thoracotomy. “(Changes in lines 45-47, page 1)
“VATS treatment of descending necrotizing mediastinitis is safe and effective, while being comparable with thoracotomy in terms of surgical outcome, morbidity, and mortality. VATS approach is associated with shorter operative time and ICU stay than thoracotomy. Early diagnosis and treatment are the key elements of DNM management. (Changes in lines 301-304, page 10)
Point 14: The bibliography is incomplete. The authors should carry out a better literature search. More recent references should be cited. At least 30 recent references are required for an original article.
Answer to the Reviewer point 14: The observation of the reviewer has been accepted and the new manuscript has been modified accordingly. We implemented the bibliography through a new literature search.
Point 15: The quality of the English language should be improved. There are many expressions, terms and sentences that need significant amount of revisions. The text would benefit from professional editing.
Answer to the Reviewer point 15: The observation of the reviewer has been accepted and the new manuscript has been modified accordingly. We revised English language of the manuscript.
Reviewer #2:
Descending necrotizing mediastinitis is one of the critical clinical diseases with a high mortality rate. The authors retrospectively analyzed treatment cases from multiple centers and compared theoutcomes of minimally invasive VATS surgery and conventional open thoracotomy surgery. They concluded that The treatment of descending necrotizing mediastinitis requires accurate pre-operative planning and multidisciplinary approach. Minimally invasive VATS approach is feasible, but if sufficient drainage cannot be achieved, thoracotomy should be performed.
This is an interesting paper, as such readable cases often come from case reports. In the current medical environment, it is difficult to encounter a large number of such diseases. For thoracic surgeons, obtaining such a conclusion is obvious. If minimally invasive surgery can solve the problem, open surgery will definitely not be chosen.
Here are still some questions that the authors need to answer or to discuss
Point 1: In Figure 2, the patient underwent left cervical incision and right thoracic surgery, but the thoracic drainage tube was clearly placed in the patient's left chest, which is unreasonable.
Answer to the Reviewer point 1: The observation of the reviewer has been accepted and the new manuscript has been modified accordingly. We made a mistake in Figure 2 legend, we revised that. The thoracic surgery was performed on the left.
“Chest CT of a 43-year-old patient showing mediastinal involvement by DNM. The patient was treated with left cervicotomy and underwent a left thoracotomy for mediastinal drainage.” (Changes in lines 132-134, page 4)
Point 2: The most important treatment for DNM lies in timely detection and early operation and drainage. Have the authors considered the relationship between the time from diagnosis to surgery and treatment outcomes?
Answer to the Reviewer point 2: Yes, we evaluated time to diagnosis and time to treatment that resulted to be very short and without significant difference between groups. We analyzed time to treatment relation with outcome also in the discussion. We thank the Reviewer for reviewing our manuscript and for commenting. The manuscript has been improved according to the suggestions of the reviewer.
“Anamnestic data, including, age, gender comorbidities, time to diagnosis, time from diagnosis to surgical treatment and mediastinitis clinical presentation were recorded.” (Changes in lines 111-112, page 3)
“The mean age of the population was 59 years and the majority of patients were male (66%). The most common source of infection was odontogenic and the most common isolated pathogen was Staphylococcus aureus (47%). The median time to diagnosis of DNM was 1 day while the median time from diagnosis to surgical treatment was 0.8 days, without significant differences between groups (p = 0.19 and p = 0.23, respectively).” (Changes in lines 209-213, page 7)
“Undoubtedly, early diagnosis and treatment is a key element in DNM management. The reported mortality rate is highly variable: while some studies report a mortality between 16 and 20% [22, 23], some others report a mortality of 30-40% [24,25]. It has been proved that early detection aided by timely diagnostic imaging is fundamental due to the rapid progression of DNM [26]. The mean time to treatment in recent literature is about 1,5 days [3, 14, 27], and it is associated with a 30- day mortality lower than 20%. In our study the median time to treatment was 0.8 days, and the 30-day mortality was 9%, hence we may assume that the rapid treatment had a favorable influence on the survival rate.” (Changes in lines 271-278, page 9)
Point 3: The surgical areas involved in mediastinal infections are not consistent, and,relatively, the choice of surgical approach may also vary greatly. Have the authors considered the feasibility of performing surgery through the subxiphoid approach in cases of bilateral VATS or cases which infection dominated in anterior mediastinum?
Answer to the Reviewer point 3: The observation of the reviewer has been accepted and the new manuscript has been modified accordingly. We have considered the use of subxiphoid approach, but in our institutions, we prefer the exposure of triportal VATS for bilateral involvement in order to explore the entire mediastinum with ease.
“The conventional surgical strategy for mediastinal drainage in patients with DNM is lateral/posterolateral thoracotomy [1,9], thanks to the optimal exposure of mediastinum, thoracic cavity and pericardium that it guarantees. Mediastinal VATS drainage has been first proposed by Roberts et al. [10] in 1997, that highlighted the possibility to achieve sufficient drainage with the decreased morbidity of a minimally invasive approach. Also, subxyphoideal thoracoscopy is a possible surgical approach, though it has some limita-tions in the posterior/lower mediastinum exploration [11]” (Changes in lines 69-75, page 2)
Point 4: 4. The authors performed simultaneous neck and chest surgeries for these cases. Is simple neck surgery feasible for type I cases in the real world? Are there any patients who undergo chest surgery because neck surgery is not enough to cure them?
Answer to the Reviewer point 4: Yes, simple neck surgery is feasible for some cases of type I DNM, bearing in mind that the infection can progress. The patient could need a reintervention on the thorax if the infection is not well controlled. Surely, patients with compromised clinical condition benefit from aggressive surgical treatment including thorax exploration and drainage.
Reviewer #3: This well written paper may be improved by:
Point 1: Adding în text and also în conclusions the rapid management of the primary site which caused the mediastinitis, besides the actual mediastinitis treatment;
Answer to the Reviewer point 1: The manuscript has been improved according to the suggestions of the reviewer. We revised the text and diagnosis clarifying the topic of early diagnosis and treatment of the primary site of infection.
“Primary infection site control, surgical debridement and broad-spectrum antibiotic therapy and early diagnosis are the key elements in mediastinitis management, that unquestionably needs to be addressed in a multidisciplinary setting [6, 8].” (Changes in lines 66-68, page 2)
“The surgical strategy was planned in relation to the primary source of infection and to cervical and mediastinal level of involvement, with the goal to drain soft tissues abscesses and to preserve the airway functionality. When needed, a bilateral approach was carried out.” (Changes in lines 154-157, page 4)
VATS treatment of descending necrotizing mediastinitis is safe and effective, while being comparable with thoracotomy in terms of surgical outcome, morbidity and mor-tality. VATS approach is associated with shorter operative time and ICU stay than thoracotomy. Early diagnosis and treatment are the key elements of DNM management. (Changes in lines 302-305, page 10)
Point 2: Adding some intraoperative images from toracoscopic mediastinal drainage – would have a hudge impact on readers;
Answer to the Reviewer point 2: The observation of the reviewer has been accepted and the new manuscript has been modified accordingly.
Point 3: Number and quality of the reference must be improved.
With these adds, I am convinced that the paper is ready to be published, in my professional opinion.
Congratulations for the authors!
Answer to the Reviewer point 3: We thank the Reviewer for reviewing our manuscript and for commenting. We have updated the bibliography with a new bibilography search.
We thank the Editor and the Reviewers for helping us to improve our paper.
The manuscript has been read and approved by all the authors.
We also declare that we have no conflict of interest in connection with this paper. We sincerely hope that the enclosed manuscript can be accepted for publication in the Journal of Clinical Medicine
Prof. C. Sagnelli
Prof. A. Fiorelli
Dr. A. Sica

Reviewer 3 Report
Comments and Suggestions for Authors
This well written paper may be improved by:
1. adding în text and also în conclusions the rapid management of the primary site which caused the mediastinitis, besides the actual mediastinitis treatment;
2. adding some intraoperative images from toracoscopic mediastinal drainage - would have a hudge impact on readers;
3. number and quality of the reference must be improved.
With these adds, I am convinced that the paper is ready to be published, in my professional opinion.
Congratulations for the authors!
Author Response

(The authors gave the same response as above.)

Round 2
Reviewer 1 Report
Comments and Suggestions for Authors
The manuscript has been significantly improved after revision, especially in terms of methodology and study design. In my opinion, the manuscript is acceptable in its present form.
Comments on the Quality of English LanguageMinor editing of English language required which can be resolved during standard editing process.
Reviewer 3 Report
Comments and Suggestions for Authors
My recommendation is to be accepted în the presentations form. Congrats!